The modulatory properties of Astragalus membranaceus treatment on endometrial cancer: an integrated pharmacological method

Zhang Qianqian
Huang Xianghua xianghuahuang0311@163.com
Department of Gynecology, Hebei Medical University Second Affiliated Hospital , Shijiazhuang , China
Módos Dezső
Electronic publication date: 2021 Aug 24
Publication date: 2021
Volume: 9
Electronic Location ID: e11995
Received 2021 Feb 23; Accepted 2021 Jul 27
Copyright: ©2021 Zhang and Xianghua
Copyright year: 2021
Copyright holder: Zhang and Xianghua
License: This is an open access article distributed under the terms of the Creative Commons Attribution License, which permits unrestricted use, distribution, reproduction and adaptation in any medium and for any purpose provided that it is properly attributed. For attribution, the original author(s), title, publication source (PeerJ) and either DOI or URL of the article must be cited.
License URL: https://creativecommons.org/licenses/by/4.0/

Keywords: Astragalus membranaceus, Network pharmacology, Endometrial cancer, Mechanism of action, Target prediction

Funding: The innovation ability of Hebei Obstetrics and Gynecology Clinical Medicine Research Center 20577705D This work was supported by the innovation ability of Hebei Obstetrics and Gynecology Clinical Medicine Research Center (20577705D). The funders had no role in study design, data collection and analysis, decision to publish, or preparation of the manuscript.

==============================
Astragalus membranaceus is a traditional Chinese medicine and has been used for adjuvant clinical therapy for a variety of cancers. However, the mechanism of its action on endometrial carcinoma is unclear. Based on the Gene Expression Omnibus (GEO) database, the Cancer Genome Atlas (TCGA) database, and the Traditional Chinese Medicine System Pharmacology Database (TCMSP™), the drug and target compounds were initially screened to construct a common network module. Twenty active compounds in Astragalus membranaceus were successfully identified, which hit by 463 potential targets related to endometrial cancer. Eight of the more highly predictive compounds (such as Jaranol, Bifendate, Isorhamnetin, Calycosin, 7-O-methylisomucronulatol, Formononetin, Kaempferol, Quercetin) were involved in DNA integrity checkpoint, cyclin-dependent protein kinase holoenzyme complex, and histone kinase activity. Additionally, Gene Ontology (GO) and Kyoto Encyclopedia of Genes and Genomes (KEGG) pathway confirmed that Astragalus membranaceus might play a role in the treatment of endometrial cancer through p53 signalling pathway, transcriptional misregulation in cancer, and endometrial cancer signalling pathway. Drug-target-pathway networks were constructed using Cytoscape to provide a visual perspective. In addition, we verified that formononetin inhibited the proliferation of endometrial cancer cells through cell viability tests and clone formation tests. And qPCR and western blot found that formononetin exerts anti-cancer effects by promoting the expression of estrogen receptor beta (ERβ) and p53. Based on a systematic network pharmacology approach, our works successfully predict the active ingredients and potential targets of Astragalus membranaceus for application to endometrial cancer and helps to illustrate mechanism of action on a comprehensive level.

Introduction

The incidence and mortality of endometrial cancer (EC) has been increasing in recent years due to obesity, diabetes mellitus, hypertension, a continuous oestrogen supply unopposed by progesterone, infertility, nulliparity, and tamoxifen use (Wang, Zhou & Wang, 2019). By 2020, there were an estimated 65,620 new cases and 12,590 deaths, making uterine cancer the fourth most common cancer in women and the fifth most common cause of cancer death (Siegel, Miller & Jemal, 2020). Although our understanding of the biology of endometrial cancer and therapeutic strategies has dramatically progressed and improved, adjuvant treatment recommendations for this cancer remain complicated and controversial (Brooks et al., 2019). The treatment includes a combination of surgery, radiation therapy, and chemotherapy, including paclitaxel and cisplatin, but these treatments may not effectively improve the prognosis and produce unwanted side effects (Colombo et al., 2016; Fontanella et al., 2017; Webb et al., 2019). Therefore, there is a strong medical need to determine more effective and safer drugs to improve the prognosis and survival of EC patients.

Astragalus membranaceus, dried root extract, also known as Astragali radix, is first recorded in the “Shen Nong’s Herbs” of the Han Dynasty 2000 years ago (Shan, Zheng & Li, 2019). So far, more than 200 compounds have been identified in Astragalus membranaceus, including saponins, polysaccharides, flavonoids, and some others. The diversity of constituents of Astragalus membranaceus is the material basis for its extensive pharmacological effects (Liao et al., 2017; Shan, Zheng & Li, 2019; Xie et al., 2016). Accumulating pharmacological studies and clinical evidence centred on Astragalus membranaceus have reported anti-inflammatory, immunomodulatory, and anti-tumour effects, including in gynecologic oncology (Auyeung, Han & Ko, 2016; Ong et al., 2019; Zhang et al., 2018). In cell models, astragaloside IV inhibits the invasion of cervical cancer by regulating various proteins on the surface of cervical cancer cells to induce autophagy (Xia, He & Cai, 2020). In addition, a recent study on myeloid Graffi tumour showed that the saponins have time- and concentration-dependent anti-cell survival and proliferation pharmacological effects (Georgieva et al., 2021). Moreover, several studies have shown that Astragalus polysaccharides inhibit the proliferation and metastasis of ovarian cancer (Guo et al., 2020a), breast cancer (Li et al., 2019), and prostate cancer (Guo et al., 2020b) through immune regulation, autophagy regulation, anti-oxidation and regulation of inflammation. However, few studies have investigated the effects of Astragalus membranaceus on EC or the biotargets and molecular mechanisms involved in the effects of Astragalus membranaceus.

Network pharmacology is an emerging and novel method to discover the basis of pharmacodynamic substances and explore their molecular mechanisms of action (Sidders et al., 2018). In particular, TCM network pharmacology focuses on the holistic and systematic nature of the interactions between TCM compounds, targets and diseases (Zhang et al., 2017). Liu et al. (2019a) used a comprehensive network pharmacology method to identify the key compounds and targets of Astragalus membranaceus for triple-negative breast cancer (TNBC). In vitro experiments showed that the effective compounds of Astragalus membranaceus might inhibit the invasion of TNBC through the PIK3CG/AKT/BCL2 pathway. Furthermore, using the same method, Feng et al. (2021) identified 28 active compounds and 61 common targets of Danggui Buxue Decoction (DBD) on metastatic colon cancer (MCC), providing a scientific reference for the treatment of MCC in traditional Chinese medicine. Therefore, TCM systems pharmacology may be a promising strategy for understanding the pharmacological targets and mechanisms of Astragalus membranaceus in EC.

In the present study, we aimed to use a network pharmacology approach to establish the effects of Astragalus membranaceus on EC and predict core targets and biological functions, pathways, and mechanisms of action. Notably, this is the first comprehensive study using multiple methods to clarify the antitumour efficacy substances and mechanism(s) of action of Astragalus membranaceus on endometrial cancer.

Materials and Methods

Data mining based on the GEO and TCGA database

The gene expression profile GSE63678 dataset (Pappa et al., 2015) was downloaded from the GEO database (https://www.ncbi.nlm.nih.gov/geo/). The platform for GSE63678 is GPL571 ([HG-U133A_2] Affymetrix Human Genome U133A 2.0 Array), including five normal tissues and seven tumour tissues. The probes were converted into the corresponding gene site according to the annotation information on the platform. All gene expression data were subjected to log2 transformation.

The Cancer Genome Atlas (TCGA) is a free and comprehensive database that includes large clusters of human cancer genome sequencing data (https://cancergenome.nih.gov/). The standard for data inclusion in our study was that the sample size must consist of both cancer and normal tissues, and that the number of normal tissue controls must be greater than or equal to three. In this study, we obtained 587 cases of gene expression data (35 cases of normal endometrial samples and 552 cases of EC samples) from TCGA official website for the Uterine Corpus Endometrial Carcinoma projects (UCEC). Before further analysis and processing, RNA sequencing data standardization was carried out in full accordance with TCGA’s release guidelines.

Identification of DEGs in EC

The “affy” R language package was applied to complete the log2 transformation, quantile normalization and median polish algorithm summarization. The probes were annotated by the Affymetrix annotation files. Microarray quality was evaluated by sample clustering based on the distance between different samples in Pearson’s correlation matrices (Liu et al., 2019b). The limma package was used to detect DEGs between the normal and EC tissues (Zhang et al., 2019). Screening criteria of —log2-fold-change (FC)— ≥ 1 and P-value < 0.05 were considered indicators of statistical significance in the identification of robust DEGs. In the same way, we used the R software (version: x64 4.1.0) and the “edgeR” package to screen the DEGs in the TCGA dataset. The screening criteria were: —log2-fold-change (FC)— ≥ 1 and P-value <0.05. Subsequently, we took the intersected DEG s of the two datasets by the Venn package in the R software.

Chemical compound database building

The Traditional Chinese Medicine System Pharmacology Database (Ru et al., 2014) (TCMSP™, https://tcmsp-e.com/) is a resource of systems pharmacology for TCMs and related compounds. We collected the data on Astragalus membranaceus active compounds using the ADME (absorption, distribution, metabolism, and excretion) filter method, whose main parameters included oral bioavailability (OB) and drug-likeness (DL) (Zhang et al., 2016). OB of ≥30% and DL of ≥0.18 were the threshold values for ADME evaluation systems. When herbal compounds met both criteria (OB ≥ 30%, DL ≥ 0.18), they were selected as candidates for further analysis. The chemical name “Astragalus membranaceus” was entered into the search box, and its pharmacokinetic properties were studied at the molecular level.

Disease targets database building and network construction

We introduced Astragalus membranaceus-regulated target genes and EC-related target genes into the Perl script to get overlapping targets and active compounds, which were the potential targets of Astragalus membranaceus against EC. Subsequently, these overlapping parts were introduced into Cytoscape 3.7.1 software (http://www.cytoscape.org/) (for network construction and visualization) (Holmås et al., 2019). In the network, “node” represents the corresponding compound and target genes, and “edge” represents the relationship between the compound and the target genes.

Protein–protein interaction (PPI) network construction and topology analysis

We introduced the overlapping gene targets into Cytoscape software for PPI analysis. The BisoGenet 3.0.0 package in the Cytoscape software was used to construct a protein interaction network (Martin et al., 2010). Then, we used the “CytoNCA” plugin of the Cytoscape software for topology analysis (Tang et al., 2015). For each node in the interaction network, “degree (DC)” and “betweenness centrality (BC)” were selected to assess its topological features. DC represents the number of node links and reflects how often a node interacts with other nodes (Missiuro et al., 2009). BC is defined as the degree to which a node is located on a path between other nodes (Raman, Damaraju & Joshi, 2014). The original PPI network was screened twice according to the values of DC and BC to obtain the core PPI network.

Gene function and pathway enrichment analysis

The “ggplot2” package in R software for Gene Ontology (GO) and Kyoto Encyclopedia of Genes and Genomes (KEGG) analysis (Maag, 2018), to obtain the biological process, cellular compounds, molecular function, and critical signalling pathways (The Gene Ontology Consortium, 2017). In addition, to better understand the complex relationship between core biological pathways and hub co-targets associated with the Astragalus membranaceus, we used Cytoscape 3.7.1 (http://www.cytoscape.org/) to build and analyse networks.

Reagents and cells

Formononetin (C16H12O4, 98% purity verified by high-performance liquid chromatography) was obtained from Phytomarker Ltd., Tianjin, China. Formononetin was dissolved in dimethyl sulfoxide (DMSO) as a stock solution of 100 mM. The 40 µM and 80 µM concentrations of formononetin in experiments were obtained by diluting the medium to a final concentration of DMSO< 0.1%(vol/vol). Cells treated with 0.1% DMSO served as a control. The solution was stored at 4 °C for further use.

Human endometrial cancer cell lines (Ishikawa, HEC-1A, and HEC-251) were purchased from the Institute of Biochemistry and Cell Biology of the Chinese Academy of Sciences (Shanghai, PR China). All cell lines were maintained in Dulbecco’s modified Eagle medium (DMEM), or DMEM/F12 supplemented with 10% heat-inactivated fetal bovine serum (FBS) and kept at a 37 °C, 5% CO2 incubator. Cell lines were routinely checked for mycoplasma contamination.

Cell viability assay

Cell proliferation was performed by a Cell Counting Kit-8 (CCK-8) (Meilun Biotechnology Co., Ltd., MA0218-5) according to the manufacturer’s protocol. Briefly, the Ishikawa, HEC-1A, and HEC-251 cell lines were trypsinized and seeded into 96-well plates at a density of 1 ×104 each well. After incubating overnight at 37 °C, the cells were treated with a concentration of 40 µM and 80 µM of formononetin for 24 h. The absorbance of each well was determined at 490 nm under a Smart Microplate Reader (USCNK, Wuhan).

Colony formation assay

The effect of formononetin on the colony-forming ability of endometrial cancer cells was evaluated by colony formation assay. The Ishikawa, HEC-1A, and HEC-251 cell lines were seeded into 6-well plates at the concentration of 400 cells per well. After incubating for 24 h at 37 °C, the cells were incubated with formononetin (40 µM and 80 µM). Three days later, most single colonies contain more than 50 cells. Cells were harvested and rinsed three times with distilled water, and images were obtained by a microscope.

RNA extraction and qPCR assays

After exposure to formononetin (40 µM and 80 µM) for 48 h, the Ishikawa, HEC-1A, and HEC-251 cells were harvested for RNA isolation by using a TRIZOL®reagent (TaKaRa, 9109). Then, cDNA was subsequently synthesized using 10 ng of RNA under standard conditions for the iScript cDNA Synthesis Kit (Fermentas Inc., USA). Quantitative polymerase chain reaction (qPCR) was performed by SYBR Premix Ex Taq II (Takara, Japan). The 2−ΔΔCt method was used to calculate the relative expression. The gene primers were listed as follows: ERβ: 5′-AATGGGGTTCTCTCCTGT’ (sense) and 5′-AGCCCAAAGTATCCCTGAC-3′(antisense); p53: 5′-CAGTAGTCAAGTAGTAACCCCTGCCTTGCACAG-3′ (sense) and 5′-CATGTAT-TACTGTGCAAGGCAGGGGTTACTACTT-3′ (antisense); Actin: 5′-CATGTACGTTGCTATCCAGGC-3′(forward) and 5′-CTCCTTAATGTCACGCACGAT-3′(reverse).

Western blotting

After 48 h of incubation with the designated concentration of formononetin, the Ishikawa, HEC-1A, and HEC-251 cells were lysed in RIPA buffer. The quantified protein was separated by 10% SDS-PAGE and subsequently transferred to polyvinylidene difluoride (PVDF) membranes (Bio-Rad, USA). Then the membrane was sealed with 5% nonfat milk and incubated with primary antibodies including ERβ (1:2500; Proteintech Group, Inc. Wuhan, China), p53 (1:800; bsm-33058M; Beijing Bioss Biological Co., Ltd, Beijing, China), and β-actin (ABclonal, Inc. Wuhan, China) for 2 h at room temperature. After three washes with the TBST solution, the samples were incubated with the appropriate secondary antibodies for 2 h. The blots were visualized with an electro-chemiluminescence (ECL) reagent kit (Beyotime, China). Image J software (National Institutes of Health, Bethesda, MD, USA) was used to quantify each protein band intensity, and the β-actin was used as an internal reference.

Statistical analysis

The data were presented as the mean ± standard deviation (SD). The SPSS 25.0 software (SPSS, Chicago, IL) was used for statistical analysis using one-way ANOVA. All experiments were carried out at least three times, and a P < 0.05 was considered significant.

Results

Endometrial cancer-related targets

Based on the DEG selection criteria of —log2-fold-change (FC)— ≥ 1 and P-value < 0.05, a total of 360 DEGs, including 204 upregulated DEGs and 156 downregulated DEGs, were identified between the normal and EC tissues after consolidation and normalization of the microarray data from the GSE63678 dataset (Fig. 1). In addition, a total of 11,531 DEGs were obtained from the TCGA database. Cluster analysis (Fig. 2) revealed 7,727 upregulated genes and 3,804 downregulated genes. The Venn package was used to screen the intersecting DEGs from both databases and generate the Venn map (Fig. 3). Finally, 268 EC-related DEGs with high reliability were obtained (Table S1 for details).

Figure 1 Volcanic map and heatmap of 360 DEGs (including 204 up-regulated genes and 156 down-regulated genes) from GSE63678.

(A) Volcano map of all genes. Pink nodes represent DEGs with logFC < 1 and P-value < 0.05. Blue nodes represent DEGs with logFC <-1 and p-values < 0.05. (B) Top 40 differentially expressed genes between normal and EC tissues. Each column represents a sample, and each row represents one gene. The gradual colour changes from blue to pink represent the gene expression changing from downregulation to upregulation. C: genes from normal tissues; T: genes from EC tissues.

Figure 2 Volcanic map and heatmap of 11531 DEGs (including 7727 up-regulated genes and 3804 down-regulated genes) from TCGA.

(A) Red and blue plots: differentially expressed mRNAs; black plots: normally expressed mRNAs. (B) Red: upregulation; blue: downregulation.

Figure 3 Venn diagram shows the intersecting DEGs from GEO and TCGA.

Blue area: GEO dataset; orange area: TCGA dataset; cross area: DEGs expressed in both databases.

Active compounds in Astragalus membranaceus and the targets

The effective compounds of Astragalus membranaceus were screened in the TCMSP database. A total of 87 chemical constituents and 953 targets of Astragalus membranaceus were searched, and 20 potentially active compounds, screened by OB > 30% and DL > 0.18, and 463 related targets of Astragalus membranaceus were selected. Table 1 shows the information on the top 10 active compounds and their targets (Table S2).

Disease targets database building and network construction

A total of 10 overlapping genes and 8 overlapping potentially active compounds were screened as all targets for Astragalus membranaceus treatment of EC (Table 2). As shown in Fig. 4, the compound-target network consisted of 18 nodes and 19 edges. The top three compounds of Astragalus membranaceus with the highest values of both DC and BC, in descending order by DC, were kaempferol (MOL000422), formononetin (MOL000392), and quercetin (MOL000098), and they will likely play a more significant role in treating EC.

Table 1 Top ten compounds in Astragalus membranaceus and the targets.

MolId	MolName	Target	
MOL000211	Mairin	Progesterone receptor	
MOL000239	Jaranol	Nitric oxide synthase, inducible	
MOL000296	Hederagenin	Progesterone receptor	
MOL000354	Isorhamnetin	Nitric oxide synthase, inducible	
MOL000371	3,9-di-O-methylnissolin	Nitric oxide synthase, inducible	
MOL000378	7-O-methylisomucronulatol	Nitric oxide synthase, inducible	
MOL000379	9,10-dimethoxypterocarpan-3-O-tail-D-glucoside	Prostaglandin G/H synthase 2	
MOL000380	(6aR,11aR)-9,10-dimethoxy-6a,11a-dihydro-6H-benzofurano[3,2-c]chromen-3-ol	Nitric oxide synthase, inducible	
MOL000387	Bifendate	Prostaglandin G/H synthase 2	
MOL000392	Formononetin	Nitric oxide synthase, inducible	

Table 2 Common target genes and corresponding active components of endometrial cancer and Astragalus membranaceus.

MolId	MolName	Common target genes	
MOL000239	Jaranol	CHEK1	
MOL000387	Bifendate	KDR	
MOL000354	Isorhamnetin	CHEK1, CCNA2	
MOL000417	Calycosin	
MOL000378	7-O-methylisomucronulatol	
MOL000392	Formononetin	
MOL000422	Kaempferol	BAX, CDK1	
MOL000098	Quercetin	BAX, RUNX1T1, CDK1, BIRC5, CCNB1, TOP2A, HK2	

Figure 4 Network of active compounds and common targets based on Cytoscape 3.7.1 (https://cytoscape.org/).

Pink ovals represent active compounds of Astragalus membranaceus. The yellow triangles in the middle represent the targets. The grey lines indicate the relationship between compounds and targets.

PPI network construction and topology analysis

We obtained the core PPI network after the original PPI network screened according to DC and BC (Fig. 5). The core PPI network consisted of 48 nodes and 528 edges. In other words, the core PPI network contained 48 compounds and target genes, and there were 528 relationships between them (Fig. 5C). Genes, namely, TP53, NPM1, YWHAZ, HSP90AA1, and BRCA1 with the highest DC and BC values, are likely the critical targets in EC.

Figure 5 The process for constructing the core PPI network of the 13 targets.

(A) The original PPI network. (B) The PPI network when DC > 61. (C) The core PPI network when DC > 61 and BC > 100.

Biological functions and pathway enrichment assays of the core targets of Astragalus membranaceus and EC

GO analysis (Figs. 6A–6C) and pathway enrichment analysis (Fig. 6D) were performed to clarify the biological functions of these 10 target genes. The screening condition was P < 0.05 and Q < 0.05. The BP category of the GO analysis results showed that the target genes were markedly associated with histone phosphorylation, DNA integrity checkpoint, and signal transduction involved in DNA integrity checkpoint. For the CC category, the target genes were markedly related to the cyclin-dependent protein kinase holoenzyme complex, condensed chromosome and serine/threonine protein kinase complex. The MF category was significantly enriched in histone kinase activity, heat shock protein binding, and Hsp70 protein binding. The KEGG signalling pathway analysis showed that the target genes played pivotal roles in the following pathways: ‘p53 signalling pathway’, ‘Transcriptional misregulation in cancer’, ‘EGFR tyrosine kinase inhibitor resistance’, and ‘Endometrial cancer’.

Figure 6 GO and KEGG pathway analyses of 13 target genes.

(A) Biological process category. (B) Cellular component category. (C) Molecular function category. (D) KEGG pathway analysis. The cut-off criterion for a significant difference was P < 0.05.

In addition, based on the target and pathway analyses, an entire compound, target and disease network was constructed by Cytoscape (v 3.7.1). As shown in Fig. 7, this compound, target, and disease interaction network had 23 nodes and 43 edges. Red inverted triangles and orange rectangles correspond to the pathway and target genes, respectively.

Figure 7 Pathway-targets analysis.

The red triangle represents the KEGG pathway. The orange square represents the related targets.

Anti-proliferation effect of formononetin on endometrial cancer cells

Formononetin, as an immunomodulator, has been widely used for centuries. Formononetin’s anti-tumour effects such as inhibiting cell proliferation, inducing cell cycle arrest and apoptosis have been confirmed in bladder cancer (Wu et al., 2017), breast cancer (Xin et al., 2019), and ovarian cancer (Park et al., 2018). However, the role of formononetin in endometrial cancer cells remains to be explored. Therefore, we choose formononetin for verification.

The effect of formononetin on the proliferation of endometrial cancer cell lines was tested by cell viability assay and clone formation assay. As shown in Fig. 8, formononetin significantly inhibited the proliferation rate of endometrial cancer cell lines compared with the control group (all P < 0.05 relative to the control group). In addition, compared with 40 µM of formononetin, higher concentrations inhibited the cells more strongly (all P < 0.05 relative to the 40 µM group). These results indicate that formononetin may inhibit cell proliferation in a dose-dependent manner.

Figure 8 Cell viability assay (A) and Colony formation assay (B) detect the effects of different concentrations of formononetin on the proliferation of endometrial cancer cell lines. *** P < 0.001.

The effect of formononetin on the expression of ERβ and p53

ERβ has been widely reported to play an important role in inhibiting cell proliferation. Furthermore, KEGG pathway analysis showed that Astragalus membranaceus could act on EC through the p53 signalling pathway. In order to explore how formononetin exerts an anti-tumor effect, we explore the differences in ERβ and p53 between different groups. In different groups of endometrial cancer cell lines, the expression of ERβ and p53 in the cells treated with formononetin was significantly increased, as expected (all P < 0.05 relative to the control group); In addition, the higher the concentration of formononetin, the more the expression of ERβ and p53 (all P < 0.05 relative to the 40 µM group; Figs. 9A and 9B). To add more evidence, we performed Western blots on ERβ and p53 using the same cells. Indeed, formononetin can promote the expression of ER β and p53 in a dose-dependent manner (Fig. 9C).

Figure 9 The qPCR assay and Western blot assay were used to explore the differences in ERβ and p53 expression when formononetin exerts an anti-tumour effect.

Relative expression of (A) ERβ and (B) p53 in endometrial cancer cell lines by qPCR assay. (C) Protein expression levels of ERβ and p53 in endometrial cancer cell lines by western blot assay. *** P < 0.001.

Discussion

Although our understanding of endometrial cancer has been profoundly advanced in recent years, the occurrence and progression of cancer is an intricate process; therefore, endometrial cancer may not effectively respond to a single therapeutic intervention measure (Arend et al., 2018; Urick & Bell, 2019). As a traditional Chinese herb, Astragalus membranaceus has been widely used clinically for the treatment of cancer, including EC. In clinical practice, Astragalus membranaceus injection can increase blood cells and immunoglobulins in patients with EC who were undergoing chemotherapy, and reduce bone marrow suppression (Xiyao, Chao & Lihong, 2019). Another clinical study showed that the expression of neutrophil gelatinase-related lipocalin and its receptor in the serum of chemotherapy patients who took Astragalus injection were reduced, and the recurrence rate within 3 years was significantly lower than that of EC patients who did not take (Yufang & Yan, 2019). Preclinical studies have shown that Astragalus injection can promote the expression of E-cadherin and reduce the expression of β-catenin in EC cells, thereby inhibiting the WNT signalling pathway to exert anti-cancer effects (Xiaoming et al., 2016). In addition, studies by Gao (2018) have shown that Astragalus injection can effectively reduce the levels of carcinoembryonic antigen 125 (CA125) and human epididymal secretory protein 4 (HE4) in the serum of EC patients, and reduce the toxicity of chemotherapy. Furthermore, regarding the Chinese medicine SH003, whose main ingredient was Astragalus membranaceus, the phase I clinical study on the safety of patients with solid tumours was also underway (Cheon & Ko, 2020). In the present study, we used a network pharmacological approach to explore the material basis and molecular mechanisms of Astragalus membranaceus for the treatment of EC.

Dong et al. (2020) studied Astragalus membranaceus against laryngeal cancer through network pharmacology, showing that quercetin, rutin, and chlorogenic acid in Astragalus membranaceus are the main compounds that exert anti-cancer effects. Research by Liu et al. (2019c) showed that 7-O-methyl iso-mucronulatol, quercetin, 3,9-di-O-methylnissolin, isorhamnetin, and formononetin in Astragalus membranaceus could activate the immune system to fight diseases by regulating the T cell immune response, the tropism of macrophages and the production of antimicrobial peptides. In the current study, we obtained 8 active compounds and 10 compound-disease targets in total. According to the analysis of the network, the interaction between these effective active compound and target genes indicates that Astragalus membranaceus acts on EC in a multi-compound and multitarget manner. The active compounds of Astragalus membranaceus with the highest DC values were kaempferol (MOL000422), quercetin (MOL000098), and formononetin (MOL000392). In addition, the top 5 target genes most related to the active compounds are CHEK1, CCNA2, KDR, BAX, and CDK. It was speculated that these targets might be key targets for Astragalus membranaceus-based treatments of EC.

Kaempferol represents one of the most common glycoside forms of aglycone flavonoids (Li et al., 2015). Previous studies have shown that kaempferol and its glycosylated derivatives have anti-inflammatory, anti-diabetic, anti-oxidant, and anti-tumour effects and have anti-cancer activity (Imran et al., 2019). High concentrations of kaempferol have a strong antitumour effect by promoting apoptosis and downregulating signalling pathways, inositol 3-kinase (PI3K)/protein kinase B (AKT) action and the epithelial-mesenchymal transition (EMT) (Chen & Chen, 2013; Wang et al., 2019). An investigation has shown that the inhibitory effect of kaempferol on endometrial cancer is induced by apoptosis, G2/M phase cell cycle arrest, inhibition of cell invasion, and the upregulation of the m-TOR/PI3K signalling pathway (Chuwa et al., 2018; Lei et al., 2019). Quercetin is one of the most important flavonoids in the human diet (Lin et al., 2017). Quercetin has attracted considerable attention because of its amazing anti-cancer properties, such as cell signalling, pro-apoptosis, anti-proliferative and anti-oxidant effects, and growth inhibition (Farooqi et al., 2018). In fact, the ability of quercetin to inhibit the action of multiple enzymes involved in cell proliferation and multiple signal transduction pathways in endometrial cancer has been established (Kaneuchi et al., 2003; Scambia et al., 1992; Wang et al., 2009).

Formononetin is widely known as a phytoestrogen that can replace oestrogen. Relevant literature and data have shown that formononetin can inhibit various cancer cells, especially cancers of the reproductive system, including breast cancer, ovarian cancer and cervical cancer (Zhang et al., 2018; Zhang, Chen & Zhang, 2019). Chen et al. found that formononetin inhibited the proliferation of ER-positive breast cancer cells and triggered apoptosis in a time and dose-dependent manner by promoting the increase of ER-β expression (Chen et al., 2013). Moreover, studies have shown that formononetin inhibits neuroinflammation by promoting the expression of ER- β (El-Bakoush & Olajide, 2018). Our network pharmacological analysis showed that formononetin, as a crucial active ingredient in Astragalus membranaceus, also played a powerful anti-cancer effect in the treatment of endometrial cancer. In vitro studies showed that formononetin inhibited the proliferation of endometrial cancer by promoting the expression of ERβ and thus exerted an anti-cancer effect. This result was consistent with previous research. The potential mechanism for the formononetin-ERβ axis to exert anti-cancer effects may be that formononetin acts on microRNA-375, RASD1, or ERα to form a negative feedback loop (Chen et al., 2018), but this requires further verification.

The target protein interaction network (PPI) analysis in our study revealed that the main target proteins of Astragalus membranaceus active compounds in treating EC are TP53, NPM1, YWHAZ, HSP90AA1, and BRCA1. Extensive research and data indicated that these target proteins were closely related to multiple cancer signalling pathways (Bian et al., 2018; Dimitrakopoulos et al., 2018; Repana et al., 2019; Suen et al., 2019; Tamura et al., 2015). TP53 was a tumour suppressor gene which could induce cell cycle arrest, apoptosis, ageing, DNA repair or metabolic changes (Stine et al., 2015). Mutations in this gene had been implicated in a variety of human cancers, including EC (Cancer Genome Atlas Research Network et al., 2013). In vitro studies showed that formononetin inhibited the proliferation of endometrial cancer by promoting the expression of p53 and thus exerted an anti-cancer effect. A recent study showed that the plant extract methoxyeugenol inhibits the proliferation of endometrial cancer cells by activating the p53/p21 signalling pathway (Costa et al., 2021).

GO function enrichment analysis showed that these co-targets were mainly in DNA damage, cell cycle, and metabolism, which were closely related to cell canceration. The KEGG gene network revealed that targets CHEK1, BAX, CDK1, CCNB1, and CCNA2 were associated with some important pathways related to cancer metabolisms, such as the p53 signalling pathway, cell cycle, cellular senescence, transcriptional misregulation in cancer, and endometrial cancer pathway. As shown in the endometrial cancer pathway (hsa05213), both BAX and BIRC5 are Astragalus membranaceus targets that can inhibit tumour proliferation (Huo et al., 2019; Sakuragi et al., 2002). These results suggest that Astragalus membranaceus can effectively cure endometrial cancer and help patients recover from this disease because it is mainly involved in cancer-related metabolism pathways.

Conclusions

This study is the first to investigate the effects and mechanisms of Astragalus membranaceus against endometrial cancer by conducting network pharmacological analysis and experimental verification. The network pharmacology analysis findings showed that kaempferol, quercetin, and formononetin from Astragalus membranaceus exert anti-endometrial cancer pharmacological effects by regulating multiple targets and pathways. The experimental validation results suggested that formononetin may play an anti-cancer effect by up-regulating the expression of ERβ and p53. The enrichment analysis of GO and KEGG showed that the anti-cancer effect of Astragalus membranaceus might be related to the regulation of cell cycle, metabolism and some important cancer-related pathways. These findings provide a scientific basis for elucidating the mechanism of action of Astragalus membranaceus in the treatment of endometrial cancer. However, since the study was based on data mining and experimental verification, further clinical validation studies should be conducted to determine the role of Astragalus membranaceus in endometrial cancer.

Supplemental Information

Supplemental Information 1 All 268 EC-related DEGs with high reliability

Click here for additional data file.

Supplemental Information 2 All compounds in Astragalus membranaceus and the targets

Click here for additional data file.

Supplemental Information 3 Raw data of WB

Click here for additional data file.

Supplemental Information 4 All DEGs–EC in TCGA dataset

Click here for additional data file.

Supplemental Information 5 Raw data and codes for active ingredient screening

Click here for additional data file.

Supplemental Information 6 Original code for differential gene screening

Click here for additional data file.

Supplemental Information 7 Raw data of p53-PCR

Click here for additional data file.

Supplemental Information 8 Raw data of PCR

Click here for additional data file.

Supplemental Information 9 Raw data of CCK8

Click here for additional data file.

Supplemental Information 10 Raw figures of Colony formation assay

Click here for additional data file.

Supplemental Information 11 The list of TCGA data set numbers

Click here for additional data file.

Additional Information and Declarations

Competing Interests

Author Contributions

Data Availability

The authors declare there are no competing interests.

Qianqian Zhang conceived and designed the experiments, performed the experiments, analyzed the data, prepared figures and/or tables, and approved the final draft.

Xianghua Huang conceived and designed the experiments, analyzed the data, authored or reviewed drafts of the paper, and approved the final draft.

The following information was supplied regarding data availability:

Raw data, including TCGA data set numbers, is available in the Supplemental Files.

The microarray-based expression data of EC patients were available at Gene Expression Omnibus: GSE63678.

The expression profile were all from the level 3 data available in the TCGA database. We used the search terms: “Primary Site” IS “Uterus” AND“Project Id ” IS “ TCGA-UCEC” AND “Workflow Type” IS “HTSeq-FPKM” AND “Data Category” IS “TranscriptomeProfiling” AND “Data Type” IS “Gene Expression Quantification.”

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
