# Peer review of "The modulatory properties of Astragalus membranaceus treatment on endometrial cancer: an integrated pharmacological method"

_PeerJ, doi:10.7717/peerj.11995_

## Round 0.1 · original submission · Major Revisions

I agree with the three reviewers' comments regarding the study. Please use more than one data source for defining DEGs, targets and compare the results with other endometrial cancer networks.

The methods were not clear so please describe them more precisely (see the reviewers' comments).

A stylistic note: please use "compounds" instead of "components".

In the network figures please use human-readable labels (not pathway or compound ids).

Due to the COVID situation doing other experimental validation can be hard, but if possible please go for it. However, describe the reasoning behind choosing a compound and targets.

Reviewer 1 ·

Basic reporting

ABSTRACT:
1. Wrong spelling of Formononetin in line 42.
2. ER-beta in line 45 should be written in full name

INTRODUCTION:
1. There is lack of coherence between paragraphs.
2. Introduction did not address what other people have done to solve the same problem, to reason why this study was conducted.
3. The introduction should include sufficient backgrounds

FIGURES
1. Overall, captions are too short and do explain the significant findings in the figures.

TABLES
1. Please be specific in the caption and avoid using word such as "some" (eg Table 1)

Experimental design

1. In many sections, please avoid defining the database/ software (eg: Cytoscape, TCMSOP, GO, KEGG etc) but straight away tell how database/ software was used in the study. For example, what materials were used to construct the network using Cytoscape.
2. What is the basis for selecting OB >30% and DL > 0.18?
3. The top 40 DEGs were mentioned once and in line 212 and did not see if these identified genes were used anywhere in the subsequent methods.
4. Methods should be described in detail so that it is reproducible by others.

Validity of the findings

RESULTS:
1. Table 1 only shows 18 examples (some are repetitive compounds). Please be specific for showing only these 18 examples.
2. Line 222-225 belongs to Materials and Methods.
3. In line 222, what is EC regulated genes and how many? How many Astralagus targets?
4. Line 234-235 belongs to Materials and Methods. Please clarify what does it by “the original PPI network was screened twice”?
5. DC & BC in line 229 are not defined.
6. How is the network validated? Any comparison to EC networks that have been published?

DISCUSSION:
1. ER-beta was not mentioned anywhere in the Materials & Methods and Discussion, why it was selected for experimental validation?
2. Discussion looks very brief and selective. For example: Why the discussion only focuses on KEGG when GO was also used.

CONCLUSION:
1. The conclusion is far-fetched which the compounds in the plants show potential for EC treatment not a promising one.
2. The aims mentioned in the Introduction were fully answered (ie mechanism of action)

Reviewer 2 ·

Basic reporting

The title, abstract, introduction, methods, results and discussion are appropriate for the content of the text. Furthermore, the article is well constructed, the bioinformatic analysis is well performed. The figures are relevant, high quality, well labelled and described.

Experimental design

The method design is original and novel. However, the research is within the scope of this journal. Research question is well defined, relevant and meaningful. The methods are highly technical, ethical and logistical. Statistical methods are chosen correctly.

Validity of the findings

All underlying data have been provided in detail. The findings are meaningful. The conclusions are well stated and relevant to the research questions. One of the disadvantages is the lack of in vitro or in vivo validation experiments.

Additional comments

This paper investigates the effect of Astragalus membranaceus on endometrial cancer. The authors identified a series of active ingredients of Astragalus membranaceus using GEO and TCMSP datasets. To explore further about the potential targets and related pathways, the authors demonstrated that Astragalus membranaceus may play a role in the treatment of endometrial cancer through p53 signalling pathway, transcriptional dysregulation in cancer, and endometrial cancer signalling pathway. Furthermore, the authors verified one of the identified components - forononetin to be able to inhibit the proliferation of endometrial cancer cells and exert anti-cancer effects by promoting the expression of ERβ. In short, this study identified the active components and targets of Astragalus membranaceus, and also suggested that Astragalus membranaceus could be considered as a potential therapy for endometrial carcinoma patients.


Editorial Criteria
BASIC REPORTING
The title, abstract, introduction, methods, results and discussion are appropriate for the content of the text. Furthermore, the article is well constructed, the bioinformatic analysis is well performed. The figures are relevant, high quality, well labelled and described.
EXPERIMENTAL DESIGN
The method design is original and novel. However, the research is within the scope of this journal. Research question is well defined, relevant and meaningful. The methods are highly technical, ethical and logistical. Statistical methods are chosen correctly.
VALIDITY OF THE FINDINGS
All underlying data have been provided in detail. The findings are meaningful. The conclusions are well stated and relevant to the research questions.
Overall, I think this paper is well written and will be of interest to the community of endometrial carcinoma pharmacogenetics and Astragalus membranaceus research in cancer. The pharmacological method and statistical analysis is valid and makes sense. The authors make it comprehensive by integrating analysis of multiple sources including GEO and TCMSP. The main strengths of this paper is that it addresses an interesting and unexplored question about the effect of Astragalus membranaceus in endometrial cancer, finds a novel discovery based on a carefully selected set of pharmacological procedures. What makes it better is the in vitro validation experiments. As such this article represents an excellent and elegant pharmacogenetics study which will almost certainly influence our thinking about the function of Astragalus membranaceus in endometrial cancer. In general, the work is convincing except some major and minor comments below:



Major Comments:

I’m wondering if there are any ongoing clinical trials focusing on any of the active ingredients identified in this study to treat endometrial carcinoma or even ovarian cancer? It will be very strong evidence for the significance of the current study if so. Please carefully search through https://clinicaltrials.gov

Considering the sample size of the GEO dataset(GSE63678) is relatively small. I recommend using the data from ICGC(International Cancer Genome Consortium) and TCGA(The Cancer Genome Atlas Program), which are national or international consortiums with much more samples. Please check the NCI Genomic Data Commons (https://portal.gdc.cancer.gov) to find more information about TCGA data and ICGC Data Portal(https://dcc.icgc.org) to find more information about ICGC data. At least, TCGA should contain Endometrial Carcinoma samples: https://www.cancer.gov/about-nci/organization/ccg/research/structural-genomics/tcga/studied-cancers/uterine

For the validation of the findings, I wonder if the CCLE dataset (Cancer Cell Line Encyclopedia https://portals.broadinstitute.org/ccle) and GDSC (Genomics of Drug Sensitivity in Cancer https://www.cancerrxgene.org/) could be used. Both CCLE and GDSC collect cell lines expression data with drug response information (eg., IC50). Please check out if any of the identified active compounds were included in CCLE or GDSC.



Minor Comments:
I would recommend adding a section for Abbreviations to include the full names for TCMSP, GO, GEO, KEGG, etc.

Please add “Cell lines were routinely checked for mycoplasma contamination.” in the Reagents and Cells section.

Annotated reviews are not available for download in order to protect the identity of reviewers who chose to remain anonymous.

Reviewer 3 ·

Basic reporting

This study explored the modulatory properties of Astragalus membranaceus treatment on endometrial cancer by using network pharmacology analysis coupled with experimental validation. The studies about the progression and treatment of endometrial cancer are important work. This study provides a scientific basis for clarifying the effects and mechanism of Astragalus against endometrial cancer. However, this article was not well organized; some of the methods and results were not shown clearly; moreover, the English writing and grammar must be extensively improved. Some major and minor concerns need to be solved before acceptance for publication, these points as discussed below:

Experimental design

no comment

Validity of the findings

no comment

Additional comments

1, The authors are encouraged to supplement in vivo experiments.
2, High quality TCM Database should be retrieved and collected in Database Construction, such as ETCM, TCMSP and TCMID database, etc., and should not be limited to the TCMSP database.
3, Please consider whether it is reasonable to use OB ≥ 30% and DL ≥ 0.18 as the judgment index.
4. The clinical application of Astragalus membranaceus against endometrial cancer should be supplemented.
5, The background of this study should be descripted in more detail in the introduction section.
6, Many active compounds were screened in the Astragalus membranaceus, why only selected formononetin to conduct the experimental validation? The activities of other active compounds need to be evaluated.
7, Why did the authors only focus on ERβ rather than other important targets?
8, Please increase the pixels of the figures, especially the figure 4.
9, Only detection of ERβ expression could not confirm the mechanism of Astragalus against endometrial cancer. More experimental validation is encouraged.
10, The role of ERβ in endometrial cancer needs to be discussed in more detail in the discussion.
11, The English writing and grammar must be extensively improved. For example, in the ABSTRACT section, the wrong spelling “forononetin” and “formonitine” should be changed to “formononetin”. The authors need to check the manuscript again to avoid some spelling mistakes.

---

## Round 0.2 · Minor Revisions

Please address the last minor things regarding Reviewer 1. After that, the manuscript should be ready for publication.

Reviewer 1 ·

Basic reporting

Overall the manuscript has improved by addressing the comments from reviewers

Experimental design

No comment

Validity of the findings

1. Figure 5: The process for constructing the core PPI network of the 13 targets

Why 13 targets when in the results say: The core PPI network consisted of 48 nodes and 528 edges.
Brief explanation for clarity will be good.

Reviewer 2 ·

Basic reporting

The title, abstract, introduction, methods, results and discussion are appropriate for the content of the text. Furthermore, the article is well constructed, the bioinformatic analysis is well performed. The figures are relevant, high quality, well labelled and described.

Experimental design

The method design is original and novel. However, the research is within the scope of this journal. Research question is well defined, relevant and meaningful. The methods are highly technical, ethical and logistical. Statistical methods are chosen correctly.

Validity of the findings

All underlying data have been provided in detail. The findings are meaningful. The conclusions are well stated and relevant to the research questions.

Additional comments

Major Comments:

I’m wondering if there are any ongoing clinical trials focusing on any of the active ingredients identified in this study to treat endometrial carcinoma or even ovarian cancer? It will be very strong evidence for the significance of the current study if so. Please carefully search through https://clinicaltrials.gov

RESPONSE: Thanks for this comment. We searched Astragalus and the selected compounds in the database. Unfortunately, we did not find any relevant clinical trials.
However, we searched the PUBMED database and found that the compound Chinese medicine of Astragalus membranaceus, Angelica gigas, and Trichosanthes kirilowii was in phase I clinical trials for the safety of solid tumour patients. And there is also some clinical trial data in China. We add these data in the Discussion section.

COMMENTS: The Discussion part looks great. Thanks!

Considering the sample size of the GEO dataset(GSE63678) is relatively small. I recommend using the data from ICGC(International Cancer Genome Consortium) and TCGA(The Cancer Genome Atlas Program), which are national or international consortiums with many more samples. Please check the NCI Genomic Data Commons (https://portal.gdc.cancer.gov) to find more information about TCGA data and ICGC Data Portal(https://dcc.icgc.org) to find more information about ICGC data. At least, TCGA should contain Endometrial Carcinoma samples: https://www.cancer.gov/about-nci/organization/ccg/research/structural-genomics/tcga/studied-cancers/uterine
RESPONSE: Thanks for this suggestion, and we concede the previous version is weak. We added the UCEC RNA-Seq data in the TCGA database and found 587 cases, including 35 normal samples and 552 tumour samples.

COMMENTS: The revised version of “Data mining based on the GEO and TCGA database” part in Methods looks good to me.

For the validation of the findings, I wonder if the CCLE dataset (Cancer Cell Line Encyclopedia https://portals.broadinstitute.org/ccle) and GDSC (Genomics of Drug Sensitivity in Cancer https://www.cancerrxgene.org/) could be used. Both CCLE and GDSC collect cell lines expression data with drug response information (eg., IC50). Please check out if any of the identified active compounds were included in CCLE or GDSC.

RESPONSE: Thanks for this comment. However, it is a pity that we searched the selected compounds in the two databases CCLE and GDSC, and found no helpful information. We guessed that these might be due to traditional Chinese medicine complexes. And we are honoured that you have provided us with two important databases to provide more guidance for our future research on drug information.

COMMENTS: It makes sense. Thanks for double checking the databases.

Minor Comments:
I would recommend adding a section for Abbreviations to include the full names for TCMSP, GO, GEO, KEGG, etc.
COMMENTS: The Abbreviations section looks good.
Please add “Cell lines were routinely checked for mycoplasma contamination.” in the Reagents and Cells section.

COMMENTS: I see the recommended sentence added. Thanks!

Annotated reviews are not available for download in order to protect the identity of reviewers who chose to remain anonymous.

Reviewer 3 ·

Basic reporting

This article predicts the active ingredients and potential targets of Astragalus membranaceus for application to endometrial cancer, which may help to illustrate mechanism of action on a comprehensive level. This manuscript has been intensively revised and improved, and could be accepted for publication after some minor revisions, for example, the words in the figure 6 are too vague to read.

Experimental design

The experimental procedures and results were clearly described.

Validity of the findings

The findings were validation.

Additional comments

This manuscript has been intensively revised and improved, and could be accepted for publication after some minor revisions, for example, the words in the figure 6 are too vague to read.

---

## Round 0.3 · accepted · Accept

The manuscript is ready for publication.